# Beyond the Syndemic of Opioid Use Disorders and HIV: The Impact of Opioids on Viral Reservoirs

**DOI:** 10.3390/v15081712

**Published:** 2023-08-09

**Authors:** Mattia Trunfio, Antoine Chaillon, Nadejda Beliakova-Bethell, Robert Deiss, Scott L. Letendre, Patricia K. Riggs, Niamh Higgins, Sara Gianella

**Affiliations:** 1Unit of Infectious Diseases, Department of Medical Sciences at Amedeo di Savoia Hospital, University of Turin, 10149 Turin, Italy; 2HIV Neurobehavioral Research Program, Department of Psychiatry, University of California San Diego (UCSD), San Diego, CA 92103, USA; 3Division of Infectious Diseases and Global Public Health, Department of Medicine, University of California San Diego (UCSD), La Jolla, CA 92037, USA; 4VA San Diego Healthcare System and Veterans Medical Research Foundation, La Jolla, CA 92037, USA; 5Department of Medicine, Owen Clinic, University of California San Diego (UCSD), San Diego, CA 92037, USA

**Keywords:** HIV, reservoir, opioids, opioid use disorders, immune system, replication

## Abstract

People with HIV are more likely to have opioid use disorder and to be prescribed opioids for chronic pain than the general population; however, the effects of opioids on the immune system and HIV persistence have not been fully elucidated. Opioids may affect HIV reservoirs during their establishment, maintenance, and reactivation by enhancing HIV infectivity and replication due to upregulation of co-receptors and impairment of innate antiviral responses. Opioids may also modulate immune cell functioning and microbial translocation and can reverse viral latency. In this review, we summarize the current findings for and against the modulating effects of opioids on HIV cellular and anatomical reservoirs, highlighting the current limitations that affect in vitro, ex vivo, and in vivo studies in the field. We propose further research targets and potential strategies to approach this topic.

## 1. Introduction

The increase in opioid prescriptions for pain in the 1990s initiated the current opioid pandemic. Heroin and synthetic opioid use along with opioid overdose deaths have more than doubled since 2000 [1]. Globally in 2020, more than 61 million people used opioids for non-medical purposes; thereby, opioid use disorder (OUD) involved about 1.2% of the global population aged 15–64 [2]. In recent years, HIV diagnoses have increased among people who inject drugs, now accounting for 12% of global infections [3]. Conversely, about 18% of people injecting opioids have HIV [4]. People with HIV (PWH) are more likely to develop OUD than the general population and to be prescribed opioids, mainly for long-term management of chronic pain; up to 75% of PWH are prescribed opioids in their lifetime [5,6,7].

OUD is currently treated with medications (MOUDs) such as agonists (methadone and buprenorphine) or antagonists (naltrexone) of opioid receptors. Their introduction in clinical practice has greatly impacted HIV incidence by reducing opioid injections, and by increasing medication adherence and retention in care [8]. Among PWH with OUD, those on MOUDs are more than twice as likely to achieve viral suppression on antiretroviral therapy (ART) than those not receiving MOUDs [8]. This effect is mostly attributable to improved engagement in care and other behavioral factors, but biological mechanisms also likely underlie the effects of opioids on the prognosis of PWH with OUD. In fact, PWH on chronic opioids have a higher risk of all-cause mortality, AIDS-related death, and progression to AIDS even after adjusting for comorbidities, access to care, and ART duration and adherence [6,9].

With the rise of the opioid public health crisis, several studies have highlighted the viral (e.g., HIV infectivity, virulence, and persistence) and immune effects of opioids, suggesting a relevant impact of opioid receptors engagement on host and viral outcomes [6,9,10,11,12]. As clinical trials for HIV cure interventions progress, it becomes crucial to comprehend the impact of opioid use on the HIV reservoir and its potential eradication. The HIV reservoir represents the primary obstacle to achieving sustained remission without the need for ART [13,14,15]. Its definition varies, encompassing the entire collection of all cells with any integrated HIV DNA; cells capable of transcription of viral RNA irrespective of genetic intactness or replication competence; or it may only include replication-competent proviruses, capable of reestablishing viremia upon ART cessation [13,14,15,16]. For the purpose of this review, the term “HIV reservoir” will predominantly adhere to the first definition, unless explicitly stated otherwise.

In this comprehensive review, we assess the existing evidence on the effects of opioids on HIV reservoir establishment and maintenance, examining findings from in vitro, ex vivo, and in vivo studies. To facilitate these efforts, we lay down four key premises that serve as an interpretive framework for investigations in this domain.

Research should account for the heterogeneity within the large class of opioid drugs. Studies aiming to assess effects of opioids on viral and immune factors should consider the pharmacokinetic and pharmacodynamic features of each distinct opioid drug, and possibly other factors that can affect cell and tissue responses to opioids, as summarized in Figure 1. For instance, during chronic opioid exposure, compensatory mechanisms and tolerance could explain contradictory findings such as reduced B-cell responses following morphine administration (acute exposure) but effective humoral immune responses to vaccination in heroin and methadone users (chronic exposure) [11]. Differences between opioid drugs may also explain discrepancies with some, but not other opioids being associated with immunosuppression [11,17,18].Research should account for the heterogeneity within the cellular and tissue-based HIV reservoirs. HIV integrates its proviral genome into the host genome, establishing a pool of infected cells that can indefinitely harbor provirus during ART [13]. This cell reservoir is heterogeneous and mainly consists of memory CD4^+^ T-cells and T-cell subsets (e.g., follicular helper, regulatory, helper-1, and helper-17 cells) [13]. More recently, evidence has accumulated on the contribution of myeloid cells, which includes macrophages and microglia, in these HIV reservoirs [19,20]. These cell types display distinct proviral transcriptional activity and inducibility, as well as different expression and activity of opioid receptors and response patterns to opioid-induced stimuli. Further, the response to opioids on similar or even the same cell types may vary quantitatively and qualitatively across tissues, which may explain why the same dose of the same opioid could differentially impact the HIV reservoir of distinct tissues such as the gut-associated lymphoid tissue and central nervous system (CNS) [10,11,21,22].Research should account for the heterogeneity in the activity of HIV reservoirs. The reservoir can be distinguished based on its replication and transcriptional activity [13]. Defective provirus genomes do not encode replication-competent viruses due to the accumulation of several mutations and deletions [13,15]. However, defective proviruses can be transcribed and can contribute to HIV pathogenesis by producing viral proteins and noncoding RNA sequences [23]. Current HIV cure strategies primarily target intact provirus genomes, which account for less than 5% of all the infected cells in peripheral blood [24,25,26,27]. Since HIV transcripts and proteins can trigger immune activation, it is important to differentiate not only intact and defective provirus but also their transcriptional capacity to understand the respective contribution to HIV pathogenesis. Furthermore, transcriptionally active reservoirs could be better targeted for clearance by host immune responses [13,15,27,28]. As discussed in this review, opioids can shape the HIV reservoir primarily through immunological effects. This observation raises the possibility that the modulation of the HIV reservoir by opioids may be more significant for transcriptionally active proviruses compared to reservoirs that undergo deep latency and complete silencing. Similar hypotheses can be made with regard to the different degrees of inducibility from latency across reservoirs [13,29].Research should address concurrent direct and indirect biological mechanisms and confounding factors. In vitro and ex vivo studies can identify molecular pathways affected by opioids, but they usually focus on single cell types and relatively short-term exposure (hours–weeks). Translation of these findings to PWH may also vary since polysubstance use is common, and exposure to contaminants of opioid drugs may also have immunomodulatory effects. In this regard, animal and human studies allow concomitant evaluation of indirect effects mediated by immune modulation and cell-to-cell interactions while accounting for relevant moderating and confounding factors, including ART. In this regard, opioids may alter the metabolism, distribution, and elimination of ART drugs via their effects on drug metabolizing enzymes and efflux transporters [30,31], although few studies have addressed these interactions to date.

## 2. Opioids and HIV Infectivity and Replication

Several studies have reported that specific opioids could enhance HIV infectivity and replication through the upregulation of HIV co-receptors and impairment of intracellular innate antiviral responses [11,32,33,34,35,36,37,38,39,40,41]. These findings may explain in vivo observations of a higher frequency of HIV founder viruses following intravenous opioid exposure and greater CD4 T-cell loss and viral set-point in macaques infected with the simian immunodeficiency virus (SIV) and undergoing long-term morphine administration [11].

Morphine, fentanyl, buprenorphine, and methadone may variably enhance CCR5 and CXCR4 expression in different cell types [11,32,33,36,37]. Methadone might even downregulate the natural chemokine competing for CCR5 (MIP-1b) in macrophages [34]. The combination of these mechanisms could facilitate HIV entry into target cells.

Since opioid receptors and CCR5/CXCR4 can undergo cross-talk with dimerization, approaches to inhibit HIV entry through bivalent compounds able to simultaneously interact with both receptors have proved promising [36]. For instance, a selective κ-opioid receptor (KOR) ligand inhibiting HIV expression in acutely infected human monocyte-derived macrophages [37], and a bivalent compound against μ-opioid receptor (MOR)-CCR5 heterodimer significantly reduced p24 levels in peripheral blood mononuclear cells (PBMCs), macrophages, and astrocytes [42]. Conversely, other studies found that such cross-interaction was polarized in susceptibility to R5-tropic HIV strains only [43], which is in line with in vitro evidence of naltrexone blocking the effect of fentanyl on the production of CCR5 but not of CXCR4 [33].

Further evidence points towards an opioid-mediated impairment in several innate and adaptive antiviral responses [35]. In vitro, methadone enhances HIV infection in a dose- and time-dependent manner through the dysregulation of antiviral factors in primary macrophages [34]: specifically, methadone favors two fundamental early steps of HIV infection (the synthesis of HIV strong-stop DNA and reverse transcription), and inhibits the expression of interferon-stimulated genes that regulate subsequent restriction factors (APOBEC3G/F and MxB) and anti-HIV miRNAs [34]. Several cellular miRNAs (miRNA-28/125b/150/223/382) target 3′UTR of HIV transcripts and induce latency, thereby inhibiting viral replication, while others (miRNA-155) can inhibit HIV entry and integration by reducing cellular factors required for viral replication [38,39]. Studies on the manipulation of cellular miRNAs as a novel approach for purging the HIV reservoir are ongoing [44], and interestingly, methadone was proved to reduce the expression of all aforementioned miRNAs [34]. The resulting effect of this methadone-induced inhibition was an increase in Gag gene expression and higher levels of p24 protein. Since pretreatment with naltrexone blocked these effects, methadone likely acts through opioid receptors in macrophages [34].

A population-based study has also reported an association between morphine use and lower expression of Toll-like receptor (TLR)-9 at the mRNA and protein levels in PBMCs of PWH [40]. Similar to miRNAs and intracellular restriction factors, TLR-9 is another innate mechanism that limits HIV replication in vitro and its therapeutic potential is under investigation in vivo [41]. Fentanyl, like morphine, can decrease the upregulation of TLR-9, which typically occurs after HIV infection, and, in neuroblastoma cells, this effect can be reversed by pretreatment with naltrexone [33]. The consequence of fentanyl-induced TLR-9 suppression was once again an increase in HIV p24 expression [33]. Figure 2 summarizes the main mechanisms by which opioids can favor HIV infectivity, replication, and reservoir establishment and maintenance.

Very recently, an in vitro study has detailed several µ-opioid receptor-mediated effects of fentanyl upon HIV-susceptible cell lines and infected lymphocytes, including CD4+ T cells from human donors, summarizing most of the previously described mechanisms [32]. Specifically, fentanyl enhanced the expression of both CXCR4 and CCR5 co-receptors in a dose-dependent manner, induced viral replication and HIV proviral DNA levels, and affected the expression of several miRNAs involved in HIV restriction pathways as well as in HIV-mediated pathogenesis. As early as after 24 h of exposure it was already capable of altering the cellular transcriptome by up- and down-regulation of several genes associated with apoptosis, antiviral/interferon response, chemokine, and NFκB signaling [32].

It is essential to conduct studies that closely replicate physiological conditions, including exposure doses, to validate the previous findings using in vivo data. These investigations will help ascertain whether opioids influence the initial viral infectivity and replication during primary HIV infection in individuals who use these drugs. Furthermore, understanding the potential impact of opioid use before or at the time of ART initiation on the size of the HIV reservoir and the relative proportions of reservoir components under viral suppression (such as intact versus defective, transcriptionally active versus deep latency) is of utmost importance.

## 3. Opioids and HIV Latency

Currently, there are limited data on the effects of opioids on HIV reservoir formation and maintenance after ART-mediated viral suppression, as well as on the effects of opioid receptor activation on HIV persistence. Chronic opioid use could predict higher viral load before starting ART and thus a potentially larger reservoir [11]. Surprisingly, lower levels of the total HIV DNA have been described in PWH after 1.5 years from ART initiation among people who inject drugs compared to other routes of HIV acquisition, suggesting that heroin and perhaps other opioids may reactivate HIV from latency [45]. This conclusion was strengthened by the fact that participants on intravenous drugs started ART after longer periods of infection (and therefore were expected to possess a larger reservoir at viral suppression) [45]. However, several other factors influence the size and activity of the reservoir during its establishment at the time of ART initiation [13,15,45,46]. Prolonged ART interruptions, which are common among PWH injecting opioids, might be the driver of higher intact proviral DNA levels [47], supporting the idea that behavioral factors resulting in repeated loss of viral suppression are more relevant. This highlights the need for performing longitudinal clinical studies to detangle biological and behavioral factors in this unique population, as well as the need to characterize better the reservoirs from a replication- and transcription-competence.

On reviewing in vitro evidence, opioids such as heroin and morphine reactivate the HIV provirus in different latently infected cell lines [48,49,50]; however, contradictory findings have also been described [17]. HIV reactivation was mostly observed at drug concentrations higher than those used for therapeutic regimens or those found in plasma of intravenous drug users [48]. At therapeutic plasma concentrations, buprenorphine, methadone, and morphine did not reactivate HIV in latently infected J-Lat 11.1 cells and U1 monocytes [17]; similarly, heroin and morphine did not increase HIV p24 expression in latently infected T lymphoblasts at micromolar or sub-micromolar concentrations [48]. Of note, at increased doses, opioid-mediated reactivation could be due to cellular necrosis, as the former was prevented by the addition of antioxidants in cell cultures [48].

The reasons for these conflicting findings may stem from doses of opioids, concomitant ART administration and efficacy, as well as most of the premises of this review, which include the type of cells and tissues and the molecular characteristics of the reservoirs assessed.

Interestingly, in ART-naïve SIV-infected rhesus macaques treated with morphine, there was no difference in cell-associated DNA/RNA levels across anatomical tissues, compared to saline-treated controls. However, among ART-suppressed macaques, morphine treatment was associated with a reduction in cell-associated DNA, intact pro-viral DNA levels, and inducible SIV reservoirs in CD4+ T-cells isolated from lymph nodes and rectum; this difference was not observed in CD4+ T-cells isolated from PBMCs, lungs, and spleen [22]. Furthermore, a concomitant increase in latent SIV myeloid reservoirs was observed in the CNS [22]. The depletion of functional SIV reservoirs in lymphoid tissues was attributed to potential morphine-induced dampening of T-cell activation; specifically, a depletion of T follicular helper cells with a shift from Tfh1 to Tfh2, an expansion of T regulatory cells. A lower level of CD4+ T- and B-cell activation was observed in morphine-treated macaques [22]. The effect of morphine on the CNS reservoir was explained by the morphine activity on CCR5 expression, the promotion of trafficking and trans-endothelial migration of peripheral leucocytes across the blood–brain barrier (for instance, due to increased monocyte–endothelial adhesion observed in other models [51]), and the previously acknowledged impairment in anti-HIV restriction factors in myeloid cells.

On the contrary, a recent ex vivo study using PBMCs from PWH on suppressive ART reported no differences in HIV intact and defective reservoir and reduced HIV-1 reactivation after stimulation through anti-CD3/CD28 beads in participants on opioids compared to those not on the drugs; this was correlated to repressed inflammatory cytokine responses (namely TNFα) [52]. However, the study had a limited sample size and high heterogeneity in the type and dose of opioids, as well as in the HIV-related characteristics of participants which may explain the discrepancy of these findings compared to other findings reported herein.

Tailored studies to describe the role of opioids on the size of HIV reservoirs according to the timing of reservoir establishment should follow. Similarly, investigations of the potential role of opioid receptor activation on T-cell chromatin and epigenetic silencing, the levels of negative and positive transcription factors, and the negative regulation of RNA processing and transport are also warranted since all these mechanisms could concur to affect reservoir formation, persistence, and reactivation.

## 4. Opioids and the Immune System

Opioids may enhance HIV replication in macrophages, PBMCs, and lymphocytes, despite some of these findings not being able to be replicated [11,17]. This discrepancy may relate to the heterogeneity in models and factors described in our premises and to the fact that specific opioids and doses may not directly alter HIV replication and latency but could indirectly shape the HIV reservoir by immune modulation in vivo.

Through direct opioid receptor interactions, different opioids can affect chemotaxis, migration, phagocytosis, apoptosis, cytokine production, proliferation, TLR expression, polarization of macrophages, T and B lymphocytes, and NK cells [10,11,12,21,53,54,55]. Manipulation of opioid receptors by partial agonists or antagonists may provide novel therapeutic options to temper opioid-related immune dysregulation, which may have secondary effects on the HIV reservoir. Indeed, not all these effects have detrimental consequences; for instance, by interfering with the CCL2-mediated transmigration of CD14+CD16+ (intermediate) monocytes across the blood–brain barrier, buprenorphine may protect the CNS from these highly activated, immune-privilege cells that carry HIV into the CNS and thereby may expand the local HIV reservoir contributing to brain injury [12].

To date, few studies have investigated the immunomodulatory effects of opioids among PWH. In ART-naïve PWH, OUD is associated with elevated markers of inflammation and immune activation, such as sCD14, IL-6, and D-dimer [56]. Among PWH on ART and with preserved CD4+ T-cell counts, OUD seems to impact the production of innate cytokines; in this population, OUD was associated with increased frequencies of intermediate and nonclassical circulating monocytes and dysregulated cytokine response to bacterial lipopolysaccharide (LPS) [57]. Interestingly, intermediate monocytes are primary producers of pro-inflammatory cytokines in response to bacterial products and monocytes that express CD16 being more susceptible to HIV infection. No data have yet led to determine whether adaptive immunity is likewise affected by opioids in PWH, but single-cell transcriptomics revealed that in chronic opioid users without HIV both T and NK lymphocytes have impaired responses to LPS [35].

Binding of opioid drugs to their receptors on immune cells is not thought to be the main driver of the chronic immune activation and inflammation that can accelerate HIV progression during long-term opioid use [58]. These phenomena can be driven indirectly by the increased trafficking of bacteria and microbial products from the gut. Several opioids can disrupt intestinal homeostasis by altering mucosal tight junctions and by decreasing the motility and causing constipation; these can in turn lead to intestinal permeability, accumulation of bacterial products and increased microbial translocation [11,58,59,60,61]. Further evidence supports the role of opioids in altering gut microbiota composition [62,63] and in bile dysregulation, which affects both gut microbiome and intestinal barrier integrity [64]. Dysbiosis, impaired intestinal wall, microbial translocation, and gut-associated myeloid activation are common features of both HIV infection and opioid use, and result in increased immune activation, systemic inflammation, and eventually immune paralysis [58,65]. While the exact reciprocal influence and contribution of HIV infection and specific opioids to this vicious cycle remains to be elucidated, one study observed that the presence of HIV infection controlled by ART did not aggravate markers of gut integrity, microbial translocation, and immune activation among people injecting heroin [59]. Compared to the effects of HIV infection alone, heroin use had a larger impact on and was independently associated with higher levels of LPS-binding protein and (1→3)-β-d-glucan, systemic inflammation, and activation of circulating monocytes and CD4+ T-cells [59].

The opioid-fueled microbial translocation and the eventual increased immune activation may also compete against ART benefits and immune reconstitution, explaining the higher immune activation in ART-suppressed PWH with OUD [57]. To date, no direct association between opioid-mediated microbial translocation and modulation of the HIV reservoir has been described. This association should be investigated as a possible contributing mechanism linked to an increased likelihood of detection of cell-associated HIV RNA and larger reservoir size [11,66].

Either morphine or hydromorphone treatment can enhance intestinal proinflammatory cytokine production and intestinal tissue damage in models of mucosal inflammation [67,68]; in addition to the aforementioned mechanisms, some opioids such as morphine can disrupt gut immune homeostasis also by inhibiting the packaging of miRNA into the extracellular vesicles of intestinal organoids secreted by crypt cells in the small and large intestine [67]. These vesicles contribute to maintaining intestinal mucosal homeostasis and are used by immune cells to fine-tune immune responses to various exogenous stimulations [67]. This example reinforces the idea that, despite well characterized anti-inflammatory and immunosuppressant properties of opioids [69], alternative effects and pathways can modulate immune response further, resulting in increased immune activation and inflammation. This paradox should also be weighted according to the premises of this review, as confounding factors (e.g., concomitant injections of proinflammatory substances) and the duration of opioid exposure are relevant sources of heterogeneity across studies reporting anti- and pro-inflammatory effects. Similarly, heterogeneity in tissue and cell responses could explain the opposite immune-modulating effects of opioids. For example, the TLR-4 signaling pathway is activated by several opioids in the CNS and is inhibited by the same drugs in peripheral immune cells [70]. The former case results in neuroinflammation in the absence of LPS stimuli, and the latter results in the peripheral suppression of LPS-induced production of the pro-inflammatory cytokines IL-6, TNFα, and IL-1b [70].

## 5. Opioids, HIV, and the Central Nervous System

The most enlightening example of the complexity of the interactions between opioids and tissue-based HIV reservoirs is represented by the CNS. This interplay is complicated by the effects of addiction on the brain and the low expression level of the CD4 receptor in the CNS. HIV has adapted to this environment and can infect glia via CD4-independent mechanisms [19], rendering the evidence of the opioid effect on HIV co-receptors expression less relevant in these cells. A large variety also exists in HIV latency and activity of CNS reservoirs, as myeloid cells, such as microglia, can be productively infected and be the source of infectious virions when ART is interrupted [20,71], while astrocytes, the most numerous cells in the brain, are restrictively infected [19]. Therefore, some of the effects of opioids in the periphery may be similar in the CNS but others may differ (e.g., latency modulation). These CNS peculiarities may explain the described opposite effects of morphine on SIV reservoirs in the CNS versus the lymphoid tissues of macaques [22].

Opioids could enhance HIV invasion, infectivity, and reservoir establishment in the CNS by several brain-specific mechanisms [10,72]; e.g., by exacerbating the release of chemo-attractants from astrocytes and altering the permeability and functioning of the blood–brain barrier, enhancing the recruitment and activation of HIV-infected myeloid cells, by increasing CCR5, CCR3, and CXCR2 expression in astrocytes, or by reducing the production of cytokines with a protective role, such as IL-8 and CCL4. Conversely, other opioids, such as buprenorphine decrease inflammation and monocyte migration into the CNS [73]. The complexity of these processes is underscored by experiments that combine opioids and ART drugs; e.g., the use of a derivative of maraviroc (CCR5-inhibitor) plus naltrexone leads to the blockade of HIV entry in astrocytes but not in microglia, while the presence of morphine inhibits the antiviral activity of maraviroc in both the cell types [10,72]. Further, in neuron–astrocyte co-cultures, CCR5 expression in astrocytes mediated the opioid-driven exacerbation of neuronal injury by Tat, but when CCR5 was pharmacologically blocked with maraviroc, morphine protected the neurons against Tat [10,72].

Another mechanism of action of opioids upon the CNS reservoir is the potential modulation of ART penetration into brain tissues. Acute morphine administration decreased intracellular concentrations of different ART drugs in astrocytes, increased their accumulation in brain microvascular endothelial cells, and had no effects on microglia or pericytes [74]. Chronic morphine exposure can increase the expression and function of drug efflux proteins, such as P-glycoprotein (P-gp, ABCB1), along the blood–brain barrier, thereby increasing substrate molecules, including some ART drugs, out of the CNS [72]. If this reduces ART drugs below therapeutic concentrations in the CNS, HIV could replicate and expand its reservoir. While plausible, no evidence that morphine-related differences in ART brain concentrations affect the CNS viral reservoir was found in ART- and morphine-treated macaques [75]. On the contrary, the detrimental effect of morphine upon CNS reservoirs was explained by an overall immune suppressive environment, altered gene expression by CNS myeloid cells, and by the activity of osteopontin, one of the morphine-induced genes [75]. Of note, the morphine-mediated decreased distribution of ART drugs and the opioid-induced recruitment of myeloid cells into the brain seem to vary by brain region [72,76], increasing the challenges of untangling the effects of opioids on the CNS reservoir. Direct measurement of ART concentration in different brain regions may better uncover the interactive effects of ART, opioids, and local HIV reservoirs.

## 6. Future Perspectives and Conclusions

To achieve HIV functional cure, the paramount goal is to identify all cellular and anatomical sanctuaries of viral reservoirs and to understand the molecular mechanisms of long-term persistence and latency-reversing interventions. The impact of opioids on reservoir seeding, persistence, and latency reversal remains understudied. Considering the substantial use of medical and recreational opioids among PWH, this knowledge gap poses a major roadblock to HIV cure. The prospects of achieving functional or sterilizing cure in HIV-positive opioid users will depend on achieving greater clarity about the interplay between opioids, the immune system, and viral reservoir dynamics before and after ART suppression.

Open questions that should be hopefully addressed soon are listed in Table 1. To provide answers, several methodological and ethical issues of potential randomized clinical trials should be considered. Ideally, measures of HIV persistence should be assessed before and after opioid administration to reduce the impact of remarkable inter-individual variability: participants should maintain effective viral suppression throughout the trial to account for the unintegrated virus; extensive cell and tissue sampling should be performed to assess differences in opioid effects according to reservoir sites, replication-capacity, and transcriptional activity of reservoirs; lastly, large-volume-blood sampling should be performed to characterize rare, infected cell types.

Interventional and tailored trials on OUDs, opioids or MOUDs, and HIV reservoirs are scarce. In the United States and Canada, there are currently nine HIV/AIDS and Substance Use Disorder cohorts following >12,000 unique participants and supported by the Collaborating Consortium of Cohorts Producing National Institute on Drug Abuse Opportunities (C3PNO). Among others, these include the AIDS Care Cohort to Evaluate Exposure to Survival Services (ACCESS; University of British Columbia, Vancouver, BC, Canada) involving PWH who use illicit drugs [77], the AIDS Linked to the IntraVenous Experience (ALIVE) Study, a community-based cohort including current and former injection drug users (John Hopkins University, Baltimore, MD, USA) [78], and the RADAR cohort study on multilevel influences on HIV and substance use in young, sexual and gender minorities (Northwestern University, Chicago, IL, USA) [79]. These cohorts have been designed to investigate mainly sociodemographic factors and clinical outcomes associated with substance use, including opioids, and they do not include viro-immunological characterizations of the opioid-reservoir relationship among the primary objectives. However, most of these cohorts have the availability of biospecimens collected over decades and include a large proportion of people without HIV to serve as controls. These cohorts have an invaluable potential to elucidate significant knowledge gaps regarding immuno-properties and virological effects of opioids, at least in peripheral blood, in a wide range of clinical settings and demographics.

A major step toward addressing these gaps, especially within the CNS, is the NIDA-supported Single Cell Opioid Responses in the Context of HIV (SCORCH) consortium. This project provides a template for leveraging data and samples from distinct cohorts for creation of a vast data resource for current and future investigators. One of the project objectives aims at uncovering cell type-specific gene expression and epigenomic changes in brain regions resulting from chronic opioid exposure to identify novel therapeutic targets to treat addiction (ongoing).

Specifically interested in HIV reservoirs, the Integrating Substance Use Treatment Research with Infectious Disease for Everyone (InSTRIDE) group at Yale School of Medicine (New Haven, CT, USA) is carrying on a prospective cohort study enrolling adult PWH that initiate MOUDs [80]. This study will provide unique data on HIV expression, proviral landscape, and clonal expansion dynamics during and after the initiation of one of the three most common MOUDs (methadone, buprenorphine, naltrexone) to inform best clinical practice and options for OUD in PWH [80].

Research projects that collect autopsy tissues and create biobanks can provide crucial data to describe opioid–reservoir interactions, overcoming part of the methodological and ethical issues. Cohort studies, such as the Last Gift cohort (LG) at the University of California San Diego (La Jolla, CA, USA), could overcome some of the challenges outlined in this review. The LG comprehensively characterizes terminally ill PWH, most of whom take prescribed opioids (see Figure 3), and who participate in HIV Cure research at the end-of-life. Participants consent to a rapid research autopsy within 6 h of death, which allows for better cell viability and nucleic acid integrity than longer postmortem times [81,82]. In this setting, the detailed and longitudinal characterization of the participants and the extensive collection of human tissues coupled with detailed characterization of the HIV reservoir across tissues (in terms of size, activity, diversity, clonality, and distribution) as well as next generation sequencing technology (single-cell and single-nucleus genomics) will allow for the molecular characterization of opioid use in the context of HIV infection and ART.

Studies based on the latest technologies may lead to the identification of novel therapeutic targets and diagnostic biomarkers aimed to unravel the impact of opioids on HIV integration and latency. Through newly developed assays that separately quantify intact and defective provirus genomes (e.g., the Intact Proviral DNA assay IPDA [26]) or that accurately characterize the relative transcriptional activity of cell reservoirs (e.g., single-cell RNA sequencing), it would be possible to investigate the hypothesis that the magnitude of opioid-induced effects varies between transcriptionally active (higher) and inactive reservoirs (lower). This data will inform the power of trials by clarifying expected effects, specific targets of anatomical and functional reservoirs, and the required length of observations.

In conclusion, understanding the effects of opioids on HIV reservoir dynamics and the impact of opioid use on the efficacy of latency reversing agents and on other approaches for eradication will help evaluate how better HIV cure strategies can be developed in PWH with and without opioid requirement.

## Figures and Tables

**Figure 1 viruses-15-01712-f001:**
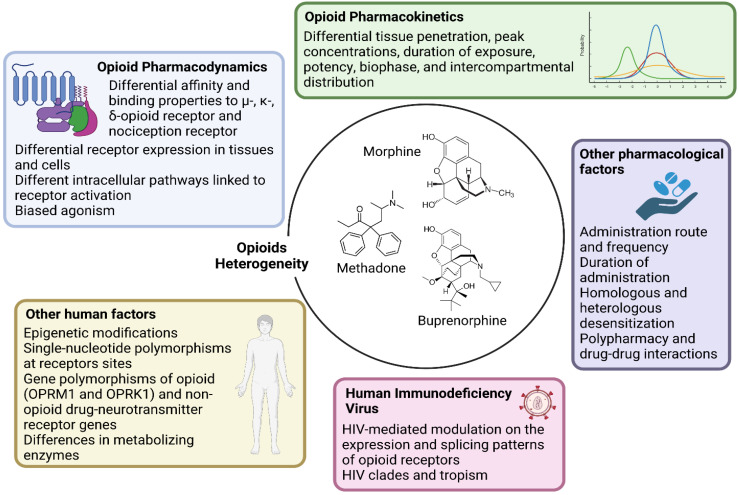
Main opioid-, human- and HIV-related factors affecting cell–opioid interactions that underlie the heterogeneity in biological responses (and thereby in scientific findings). Each opioid molecule and the three classical G-protein-coupled opioid receptors (the µ-MOR-, κ-KOR-, and δ-opioid receptor) plus the fourth nociception receptor largely vary in terms of affinity and binding patterns. These receptors are differentially expressed in several cell types and activate distinct intracellular pathways. Being full agonist (such as morphine, fentanyl, and methadone), partial agonist (such as buprenorphine) or antagonist also implies different binding properties that facilitate unique intracellular signaling. Large intra- and inter-individual variation in response to the same dose and molecule exists depending also on epigenetic modifications and single-nucleotide polymorphisms at receptors sites or in enzymes involved in opioid metabolism. Further, pharmacokinetic variability and differential tissue penetration lead to large variation in peak concentrations, duration of exposure, potency, biophase, and intercompartmental distribution, which are also dependent on the route of administration. Opioid receptors differentially undergo homologous and heterologous desensitization, which can decrease the responses to opioid agonists according to dose and length of exposure. Lastly, HIV itself could affect the expression and splicing pattern of opioid receptors.

**Figure 2 viruses-15-01712-f002:**
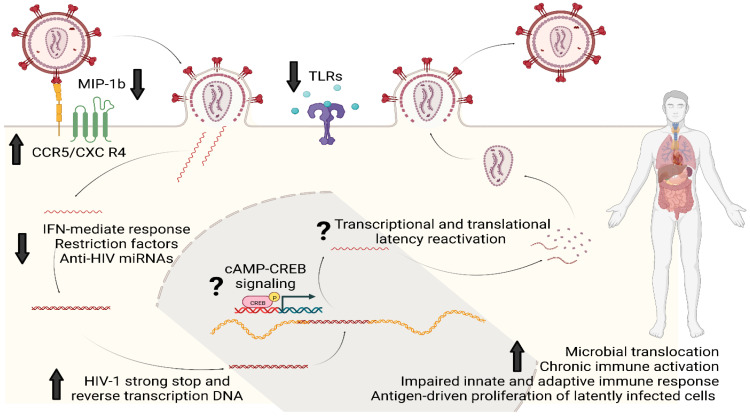
Opioids and HIV reservoir: summary of the main molecular mechanisms of interaction. Opioids could enhance HIV infectivity and replication through upregulation of HIV co-receptors (CCR5 and CXCR4) and by downregulating MIP-1b, the natural chemokine competing for CCR5. Opioids can also impair several intracellular innate antiviral responses based on IFN genes, restriction factors such as APOBEC3G/F and MxB, anti-HIV miRNAs and the intracellular pathways activated by TLRs; this would eventually result in favoring the synthesis of HIV strong-stop DNA and reverse transcription DNA as well as replication and integration. Chronic opioid exposure could induce cAMP production: increased intracellular levels of cAMP can activate proteins such as CREB which eventually could bind to the 5′LTR of HIV genome to stimulate viral replication. Lastly, opioids can increase the systemic and tissue-specific inflammatory milieu by immunomodulating properties and by worsening microbial translocation. This immuno-asset could eventually favor HIV reservoir maintenance and expansion by antigen-driven proliferation of latently infected cells and clonal expansion. Legend: MIP-1b, Macrophage Inflammatory Proteins 1 beta; CCR5, Cysteine–Cysteine Chemokine Receptor 5; CXCR4, C-X-C motif chemokine receptor type 4; IFN, interferon; miRNAs, micro RNAs; cAMP, Cyclic adenosine monophosphate; CREB, cAMP-response element binding protein cellular transcription factor; TLRs, Toll-like receptors.

**Figure 3 viruses-15-01712-f003:**
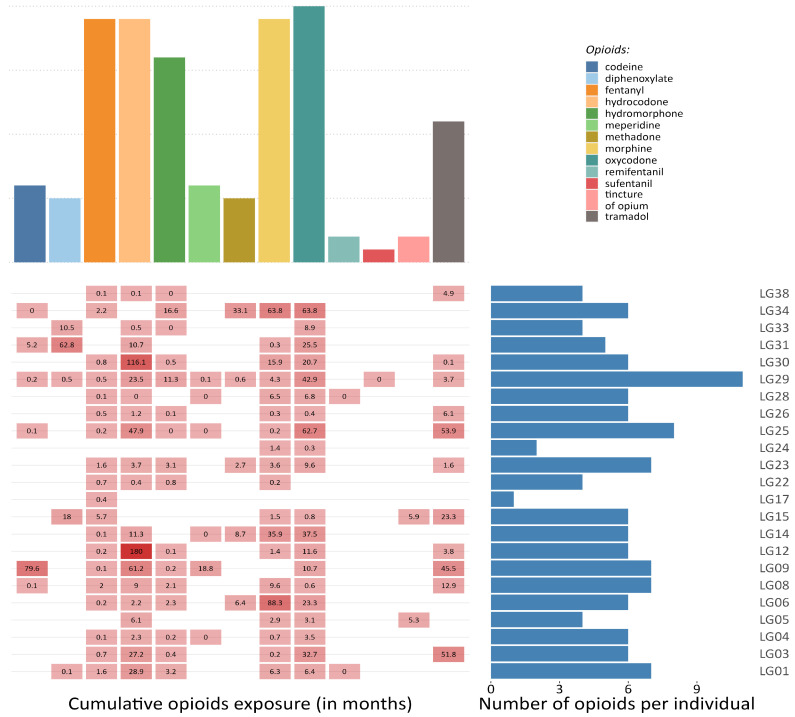
Opioid exposure in a sample of participants from the Last Gift project. The figure illustrates the frequency of different prescribed opioids among a sample of 23 participants from the Last Gift projects (**upper** panel), the cumulative exposure (in months) to each medical opioid, and the overall number of opioids that have been prescribed per individual during the last 20 years (**lower** panel).

**Table 1 viruses-15-01712-t001:** Main research questions to be addressed to disentangle the interplay between HIV reservoir, immune system, and opioids.

Research Question	Clinical and Experimental Implications
Which are the exact regulatory mechanisms in promoting reservoir establishment, maintenance, residual transcription, activity, and reactivation and how does each opioid affect these mechanisms according to dose and in vivo concentrations, route of administration, frequency, and duration of exposure?	Opioid/opioid receptor-based strategies to promote HIV reservoir restraint, depletion, or latency reversal. Targeted pharmacological interventions to support eradication strategies involving people with HIV on opioids. Pharmacological prescribing/de-prescribing to target specific body sanctuaries, such as gut and the central nervous system. Tailored selection of medical opioids with the lowest or null potential effect on HIV reservoir and immune system.
What are the differences among the available options of medications for opioid use disorder and among the analgesic opioid alternatives in terms of modulation of pro-viral landscape and the immune system?	
Is there any interaction between opioids, antiretroviral regimens and other concurrent medications/substances in modulating HIV reservoir and the immune system?	
Does HIV itself modulate host tissues and cell sensitivity and responses to opioids?	
What are the differences in reservoir establishment, maintenance, and reactivation among cell- and tissue-based HIV reservoirs and how does each opioid affect these mechanisms in each cell and body compartment?	

## Data Availability

Not applicable.

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
