# Peer review of "Beyond the Syndemic of Opioid Use Disorders and HIV: The Impact of Opioids on Viral Reservoirs"

_viruses, 2023, doi:10.3390/v15081712_

Round 1
Reviewer 1 Report
This manuscript is very puzzling, although I fully agree with its main message and its potential impact. The main message is: " we do not have a clue about the viral reservoir of the PWID. " However to reach this conclusion, authors divide the arguments into 4 main topics addressing viral infection, latency, immune system and CNS. For all these four, literature review tried to compare apples and oranges, such as morphine and heroin, macaques and humans, HIV infection before drug use and its reverse, … The bottom line conclusion remains : we don’t know.
The end of the manuscript however open up a framework for addressing the main question. This aspect is clearly of interest for the readers with the opioid exposure score too shortly developed. It will be important to start the manuscript with a clear definition of the viral reservoirs including the notion of defective and intact viral reservoirs that is never mentioned. This could be an axis of the framework, together with HIV infection timing, polysubstance use, ...
Additional comments:
L90-93: not only quantitatively but also qualitatively
L109: need a reference
L355: The LG cohort is extremely interesting in this respect, but other collections will be useful in answering most of the questions that remain. This can be mentioned to encourage others to generate and accumulate data.
Author Response
Dear Editors,
Dr. Marcondes and Dr. Kaul,
Many thanks for the detailed review of our manuscript.
We are grateful for the opportunity to resubmit a revised manuscript, and we have revised it in accordance with the Reviewers’ insightful and detailed comments.
We believe that the peer review has resulted in a much stronger manuscript which we hope is now suitable for publication.
Please find our point-by-point responses to the Reviewers’ comments below.
Many thanks for considering our manuscript for Viruses.
Sincerely,
Mattia Trunfio,
on behalf of the authors
------------------
Guest editor’s comment: State a clear definition of the viral reservoirs including the notion of defective and intact viral reservoirs in the introduction of the manuscript.
AR: As suggested, we have included a clear definition of the HIV reservoir (lines 53-62 of the introduction). Additionally, to address similar concerns raised by Reviewer#1, we have included a paragraph discussing the needs to investigate the relationship between opioid use and reservoirs measures in terms of intactness and transcriptional activity (lines 96-113).
--------------
R1: This manuscript is very puzzling, although I fully agree with its main message and its potential impact. The main message is: " we do not have a clue about the viral reservoir of the PWID. " However, to reach this conclusion, authors divide the arguments into 4 main topics addressing viral infection, latency, immune system, and CNS. For all these four, literature review tried to compare apples and oranges, such as morphine and heroin, macaques and humans, HIV infection before drug use and its reverse, … The bottom-line conclusion remains: we don’t know.
The end of the manuscript however opens up a framework for addressing the main question. This aspect is clearly of interest for the readers with the opioid exposure score too shortly developed. It will be important to start the manuscript with a clear definition of the viral reservoirs including the notion of defective and intact viral reservoirs that is never mentioned. This could be an axis of the framework, together with HIV infection timing, polysubstance use, ...
AR: We thank Reviewer#1 for their insightful review and for recognizing the complexity of this topic. As mentioned above, we have included a new paragraph to discuss the importance of distinguishing intact from defective proviral reservoirs and their transcriptional activity (see lines 96-113). These concepts are also included in several passages of the manuscript (see lines 221-224, 241-243, 256, 374-377, 431-432, and 489-496).
We agree that the topic of opioid use and its impact on HIV persistence is complex, understudied and by nature subject to multiple confounding factors. In this review, we took a descriptive approach, presenting an exhaustive review of the literature with the goal of generating provocative hypothesis and point to future directions, rather than drawing general conclusions, which might not be supported by available data. A more focused approach, e.g., only including morphine data, or human studies would have led to a partial and less informative framework. In fact, we start our review presenting 4 “premises” to highlight this heterogeneity as the critical framework for all subsequent discussion. The subsequent choice of topics attempted to organize the sparse available evidence, by recollecting the findings into three main mechanisms (infection/replication capacity, immunity, and latency reversal) and then making an example based on tissue/compartment reservoir (CNS).
Additional comments:
L90-93: not only quantitatively but also qualitatively
AR: Thank you, we have now specified as suggested (lines 91-92)
L109: need a reference
AR: We have reported the references in support of the sentence after a second review of the literature to include the most recent publications that followed our first submission (see line 141).
L355: The LG cohort is extremely interesting in this respect, but other collections will be useful in answering most of the questions that remain. This can be mentioned to encourage others to generate and accumulate data.
AR: Thank you for the appreciation and suggestion. We certainly agree and also hope this review inspires others to gather the needed data to address the remaining knowledge gaps. In this revision, we have included the examples of ongoing research efforts in this area (e.g., SCORCH) and cohort studies focusing on substance abuse in PWH supported by the C3PNO (Collaborating Consortium of Cohorts Producing NIDA Opportunities) that have data, resources, and expertise to address many of the research questions raised in this reviewed (lines 438-465). We have also expanded the description of the cohort study (already cited in the previous version) on the effects of three different medications for opioid use disorders on HIV reservoirs dynamics (lines 466-472).
Thanks to Reviewer #1 for their time spent at improving our manuscript.

Reviewer 2 Report
This is an informative and a nicely written review focusing on a specific topic
Major Comments:
- The statement "Several studies have reported that specific opioids could enhance HIV infectivity and replication through upregulation of HIV co-receptors and impairment of intracellular innate antiviral responses" should be properly referenced to support its validity.
- It is important to provide references and engage in a more comprehensive discussion regarding the claim that "opioids can increase the systemic and tissue-specific inflammatory milieu by immune modulating properties and by worsening microbial translocation," as the anti-inflammatory properties of opioids are well-established.
- It would be beneficial to include a discussion on the impact of opioids on inflammation and their ability to suppress cell metabolism/activation. This inclusion will contribute to a more comprehensive understanding of the topic at hand.
Minor Comments:
- The review is already clear in its current form, but some minor English editing would further enhance its impact. Specifically, a few sentences should be rephrased to provide clearer meaning.
- Some minor English editing would further enhance its impact. Specifically, a few sentences should be rephrased to provide clearer meaning.
Author Response
Dear Editors,
Dr. Marcondes and Dr. Kaul,
Many thanks for the detailed review of our manuscript.
We are grateful for the opportunity to resubmit a revised manuscript, and we have revised it in accordance with the Reviewers’ insightful and detailed comments.
We believe that the peer review has resulted in a much stronger manuscript which we hope is now suitable for publication.
Please find our point-by-point responses to the Reviewers’ comments below.
Many thanks for considering our manuscript for Viruses.
Sincerely,
Mattia Trunfio,
on behalf of the authors
------------------
Guest editor’s comment: State a clear definition of the viral reservoirs including the notion of defective and intact viral reservoirs in the introduction of the manuscript.
AR: As suggested, we have included a clear definition of the HIV reservoir (lines 53-62 of the introduction). Additionally, to address similar concerns raised by Reviewer#1, we have included a paragraph discussing the needs to investigate the relationship between opioid use and reservoirs measures in terms of intactness and transcriptional activity (lines 96-113).
--------------
R2: This is an informative and a nicely written review focusing on a specific topic
Major Comments:
- The statement "Several studies have reported that specific opioids could enhance HIV infectivity and replication through upregulation of HIV co-receptors and impairment of intracellular innate antiviral responses" should be properly referenced to support its validity.
AR: We have now reported the references in support of the sentence after a second review of the literature to include the most recent publications that followed our first submission (line 141).
- It is important to provide references and engage in a more comprehensive discussion regarding the claim that "opioids can increase the systemic and tissue-specific inflammatory milieu by immune modulating properties and by worsening microbial translocation," as the anti-inflammatory properties of opioids are well-established.
- It would be beneficial to include a discussion on the impact of opioids on inflammation and their ability to suppress cell metabolism/activation. This inclusion will contribute to a more comprehensive understanding of the topic at hand.
AR to R2 comments 2 and 3: We have reported more references in support of the sentence highlighted by Reviewer#2, which has been revised and further detailed (lines 318-330; references for the specific topic 11, 58-69); we have also expanded with two more examples the discussion on opioids/inflammation/immuno-modulation and activation, to better clarify the complexity of the apparent paradox for which studies have attributed both anti- and pro-inflammatory effects to opioids (lines 345-365).
Minor Comments:
- The review is already clear in its current form, but some minor English editing would further enhance its impact. Specifically, a few sentences should be rephrased to provide clearer meaning.
AR: thank you, we have reviewed English and typos throughout the manuscript. Thanks to Reviewer #2 for their time spent at improving our manuscript.

Reviewer 3 Report
Great review with the inclusion of new data using different opiods. The English needs some work specially several sentences are repetitive or need some synonymous
example
line 87, ...different cell types display different ....there several repetitions like this
inconclusive words line 331
But overall a pleasure to read
Minor changes are necessary
Author Response
Dear Editors,
Dr. Marcondes and Dr. Kaul,
Many thanks for the detailed review of our manuscript.
We are grateful for the opportunity to resubmit a revised manuscript, and we have revised it in accordance with the Reviewers’ insightful and detailed comments.
We believe that the peer review has resulted in a much stronger manuscript which we hope is now suitable for publication.
Please find our point-by-point responses to the Reviewers’ comments below.
Many thanks for considering our manuscript for Viruses.
Sincerely,
Mattia Trunfio,
on behalf of the authors
------------------
Guest editor’s comment: State a clear definition of the viral reservoirs including the notion of defective and intact viral reservoirs in the introduction of the manuscript.
AR: As suggested, we have included a clear definition of the HIV reservoir (lines 53-62 of the introduction). Additionally, to address similar concerns raised by Reviewer#1, we have included a paragraph discussing the needs to investigate the relationship between opioid use and reservoirs measures in terms of intactness and transcriptional activity (lines 96-113).
--------------
R3: Great review with the inclusion of new data using different opiods. The English needs some work specially several sentences are repetitive or need some synonymous
example line 87, ...different cell types display different ....there several repetitions like this inconclusive words line 331
But overall a pleasure to read
AR: thanks to Reviewer #3 for their appreciation and time spent on our manuscript; we have reviewed English and typos throughout the manuscript.
We have also updated Figure 3 with more data from the ongoing collection and characterization of the opioid intake from more participants of the Last Gift cohort.

Round 2
Reviewer 1 Report
I was pleased to read this revised version and I recommend its publication